# Benefits of Combined Phage–Antibiotic Therapy for the Control of Antibiotic-Resistant Bacteria: A Literature Review

**DOI:** 10.3390/antibiotics11070839

**Published:** 2022-06-22

**Authors:** Kevin Diallo, Alain Dublanchet

**Affiliations:** 1Department of Infective and Tropical Diseases and Internal Medicine, University Hospital of la Reunion, 97448 Saint-Pierre, France; 22465 Rue Céline Robert, 94300 Vincennes, France; adublanchet@orange.fr

**Keywords:** phage therapy, antibiotic therapy, bacteriophage, synergy, combined activity

## Abstract

With the increase in bacterial resistance to antibiotics, more and more therapeutic failures are being reported worldwide. The market for antibiotics is now broken due to the high cost of developing new molecules. A promising solution to bacterial resistance is combined phage–antibiotic therapy, a century-old method that can potentiate existing antibiotics by prolonging or even restoring their activity against specific bacteria. The aim of this literature review was to provide an overview of different phage–antibiotic combinations and to describe the possible mechanisms of phage–antibiotic synergy.

## 1. Introduction

The use and overuse of antibiotics over the past 90 years has led to the inexorable development of bacterial resistance, with more and more therapeutic failures being reported worldwide [1]. Until the early 21st century, this problem was compounded by the belief that bacterial resistance could be overcome with the commercialisation of new antibiotics. Yet, in the last two decades, the market for antibiotics has been broken due to the high cost of developing new molecules. As the hope of repeating the therapeutic breakthroughs of the 20th century is rapidly fading away, there is an urgent need to develop novel approaches for the control of antibiotic-resistant bacteria [2].

While solutions to the problem of bacterial resistance are emerging, the most innovative are still in development and will require time to become clinically operational. This situation has renewed interest in phage therapy, a century-old method whose efficacy has been well-described in the literature.

Phages (or bacteriophages) are bacteria-specific viruses that replicate by using the metabolism of bacteria [3]. They are the most abundant microorganisms in the biosphere and are present in all areas where bacteria grow. Insofar as these microorganisms help to regulate bacterial populations, they contribute to the homeostasis of ecosystems. Phages were first characterised as ultra-microscopic viruses by Frederick Twort in 1915 and independently described as bacteriophages by Félix d’Hérelle in 1917 [4,5].

The majority of phages are composed of a DNA or RNA genome packaged inside a protein shell (capsid) and a tail that recognises the bacterial host. When a phage detects a bacterial receptor, it irreversibly adsorbs to the host bacterial cell and injects its genome into it, with the capsid remaining outside [6]. In the lytic cycle, the injected genome replicates and kills the infected cell, thereby inhibiting the growth of the bacterial population. In the lysogenic cycle, the injected genome remains in the host cell without replicating. The present literature review focuses on lytic phages, as only these can kill the bacteria they infect [3].

Phages have several advantages: their ability to replicate can lead to self-amplification; they have greater specificity than antibiotics; they have a low rate of side effects; they can be used to treat patients who are allergic to antibiotics; their production costs are low; and they can be administered via different routes [7].

Phages are especially effective against bacteria when combined with antibiotics. This phenomenon, known as phage–antibiotic synergy (PAS), is explained by the fact that the administration of sub-lethal doses of antibiotics in phage-infected bacteria significantly increases the size of phage plaques [8]. Most interestingly, the combination of phages with existing antibiotics can potentiate the latter by prolonging or even restoring their activity against specific bacteria.

The aim of this literature review was to provide an overview of different phage–antibiotic combinations and to describe the possible mechanisms of PAS.

## 2. Historical Development of Combined Phage–Antibiotic Therapy

The first half of the 20th century saw the emergence and development of phage therapy for the treatment of infectious diseases. Following the discovery of penicillin in 1928, phages were rapidly replaced with antibiotics in clinical practice because the mechanism of action of antibiotics proved easier to understand. From the 1940s onwards, different therapeutic combinations of phages and antibiotics were evaluated in several studies.

### 2.1. Combination of Phages with Sulphonamides

In the early 20th century, phages were often used in association with sulphonamides to treat infectious diseases. The synergistic effect of this combination was first demonstrated in 1941, when Zaytzeff-Jern et al. showed that combining phages with sulphanilamide and sulphapyridine inhibited the in vivo growth of *Staphylococcus aureus* and *Escherichia coli* [9]. That same year, Neter et al. reported that phages completely recovered their antibacterial activity in broth cultures of phage-resistant *Staphylococci* strains when combined with sulphanilamide, sulphapyridine and sulphathiazole [10]. In 1942, MacNeal et al. compared the effectiveness of phage therapy to that of combined phage–sulphathiazole therapy in 56 rabbits with staphylococcal meningitis. The superiority of phage–sulphathiazole therapy was striking. All 23 rabbits that did not receive treatment died; all 5 rabbits that received a late combination of sulphathiazole and phages died; 1 rabbit survived out of the 6 that received early sulphathiazole treatment; and 13 rabbits survived out of the 22 that received an early combination of sulphathiazole and phages [11]. In the 1943 study by MacNeal et al., the joint administration of phages and sulphonamides also proved effective in humans, as 24 out of 55 patients treated for acute malignant staphylococcal infections were cured [12,13].

### 2.2. Combination of Phages with Penicillin

Following the discovery of penicillin in 1928, a number of researchers evaluated the effectiveness of combining this antibiotic with phages for the treatment of bacterial infections. In 1945, Himmelweit et al. broth cultured 130 million *Staphylococci*/mL, to which they added (a) penicillin, (b) phages and (c) a combination of both. They found that complete lysis occurred in 6 h and 3 min for penicillin alone (a), in 1 h and 55 min for phages alone (b) and in 1 h and 25 min for the phage–penicillin combination (c) [14]. In 1945–1946, MacNeal et al. reported cases of successful phage–penicillin therapy for the following indications: endocarditis [15,16], bacteraemia, osteomyelitis and peritonitis [17].

## 3. Examples of Synergistic Phage–Antibiotic Combinations

In their literature review of studies evaluating phage–antibiotic combinations, Tagliaferri et al. noted that the vast majority of combinations showed favourable results [18]. Several other studies have described synergistic phage–antibiotic combinations against specific bacteria.

Examples of synergistic combinations are described below and summarised in Table 1. We considered these combinations to be synergistic in vitro if they were reported to be more effective than phage therapy alone or antibiotic therapy alone. We considered them to be synergistic in vivo if treatment was reported to be a clinical success, even though their efficacy could not be explicitly compared to that of either therapy alone.

### 3.1. Synergistic Phage–Antibiotic Combinations against Staphylococcus aureus

In their 2012 study, Kirby et al. described the synergistic action of phage–gentamicin combinations against *S. aureus* [37]. In 2015, Ali et al. confirmed that adding phages to gentamicin eradicates *S. aureus* strains and observed the same phenomenon with vancomycin and tetracycline [46]. Chhibber et al. highlighted the synergistic effect of combining phages with linezolid against methicillin-resistant strains of *S. aureus* [39].

### 3.2. Synergistic Phage–Antibiotic Combinations against Pseudomonas aeruginosa

Over the past twenty years, several studies have described phage–antibiotic combinations with PAS against *Pseudomonas aeruginosa*.

The study by Hagens et al. reported a significant increase in in vitro sensitivity to several antibiotics in two *P. aeruginosa* strains infected with the filamentous phages Pf3 and Pf1 [31]. Knezevic et al. found that the combination of phages with subinhibitory concentrations of ceftriaxone reduced the growth of *P. aeruginosa* [40]. The study by Torres-Barceló et al. showed that combining phages and antibiotics substantially increased bacterial control compared to using either separately [45]. In their study evaluating the combination of phage PEV20 with five inhaled antibiotics (ciprofloxacin, tobramycin, colistin, aztreonam and amikacin) against *P. aeruginosa* infections, Lin et al. noted that the association of phage PEV20 with ciprofloxacin exhibited the strongest synergistic effect [50]. In their evaluation of different phage–antibiotic combinations against *P. aeruginosa*, Uchiyama et al. found that combinations with piperacillin and ceftazidime had the strongest PAS [51].

### 3.3. Other Synergistic Phage–Antibiotic Combinations

Several cases of infection with Gram-negative bacteria treated with phages and antibiotics have been reported in the literature [29,32,38]. In 2004, Huff et al. assessed the therapeutic efficacy of using phages and enrofloxacin separately and in combination for the treatment of *E. coli* infections; they concluded that the phage–enrofloxacin combination was the most effective treatment [63]. In their 2020 study, Weber et al. showed the utility of adding phage SALSA to ampicillin/sulbactam for the control of intrinsic *Serratia marcescens* resistance [59]. Other Gram-negative bacteria with intrinsic resistance to antibiotics, namely *Burkholderia cepacia*, *Achromobacter* spp. and *Acinetobacter baumannii*, were shown to be susceptible to PAS [47,56,57].

In 2020, Morrisette et al. [58] found the combination of phages and daptomycin to be effective in vitro against *Enterococcus faecium*. In 2014, Cai et al. described the in vitro synergistic effect of combining ethambutol with phage D29 against *Mycobacterium smegmatis* [42].

## 4. Examples of Antagonistic Phage–Antibiotic Combinations

Antagonistic phage–antibiotic combinations have also been described in the literature. Examples of these combinations are provided below and summarised in Table 1. We considered these combinations to be antagonistic in vitro if they were reported to be less effective than phage therapy alone or antibiotic therapy alone.

In 1945, Jones et al. found that streptothricin, streptomycin and clavacin caused inactivation of different phages in bacterial cultures [64]. In 1949, Edlinger et al. and Faguet et al. observed that some bacteria developed resistance to phages in the presence of streptomycin [19,20,21,22], chloromycetin and aureomycin [23,24] and terramycin [25]. In their 1951 study, Hall et al. tested 45 antibiotics, 8 of which showed anti-phage activity: quinones (3 molecules), clavacin, subtilin, penicillic acid, polymyxin B and streptomycin [26]. In 1954, Kirby et al. noted that combining phages with aureomycin, oxytetracycline or chloramphenicol slowed the rate of killing of *S. aureus* [27].

Two decades later, Naimski et al. reported that rifampicin considerably lost its antibacterial power when associated in vitro with phage f2. They explained this by the fact that the phage capsid acts as a barrier that prevents rifampicin from binding to the usual bacterial sites [30]. More recently, Zuo et al. observed antagonism between phages and aminoglycosides [60].

Kortright et al. suggested that whether the interaction between phages and antibiotics is synergistic or antagonistic depends on the mechanism of bacterial inhibition [65].

## 5. Examples of Clinical Indications for Combined Phage–Antibiotic Therapy

Researchers have described several clinical indications for combined phage–antibiotic therapy.

### 5.1. Treatment of Infectious Diseases

In 2019, Kortright et al. reviewed studies evaluating the in vivo efficacy of phage–antibiotic combinations against various types of bacterial infection [65]. Most of these studies reported favourable clinical results.

### 5.2. Eradication of Bacterial Colonisation

In 1965, Zilisteanu et al. reported the successful eradication of *Dysenteric bacilli* colonisation in 69 out of 71 patients treated with a combination of phages and antibiotics compared to 15 out of 18 patients treated with antibiotics alone [28].

### 5.3. Eradication of Bacterial Biofilm

Over the past decade, there has been an increase in prosthetic material infections worldwide. As bacterial biofilms are especially difficult to eradicate, few antibiotics are effective against them when used alone. Nevertheless, studies have suggested that adding phages to antibiotics increases the latter’s effectiveness against biofilm of *Salmonella enterica* [34], *Klebsiella pneumoniae* [49], *P. aeruginosa* [33,35], *S. aureus* [36], *S. aureus* and *P. aeruginosa* [41] and *E. coli* and *P. aeruginosa* [43]. In addition, Kaur et al. confirmed the in vitro and in vivo efficacy of combining linezolid with phages to treat osteoarticular *S. aureus* infections linked to orthopaedic devices [48,49]. Lastly, Shlezinger et al. successfully eradicated vancomycin-resistant *Enterococcus faecalis* biofilm with a combination of phages and vancomycin [53].

### 5.4. Compassionate Use in the French Context

In France, combined phage–antibiotic therapy is highly regulated and used only for the purpose of compassionate care. In their 2020 case study, Ferry et al. showed the feasibility of adding phages to a hydrogel to treat a patient with a megaprosthesis infection secondary to orthopaedic surgery [55]. Despite an initially favourable microbiological outcome, the patient eventually underwent amputation (which led us to classify this study as a clinical failure in Table 1). In 2020, Bleibtreu et al. reported a salvage therapy strategy for a complex *S. aureus* extradural empyema that consisted of combining parenteral dalbavancin with the local administration of two different types of phages through the skin’s fistula [54].

## 6. Sequence of Phage and Antibiotic Administration

In the 1940s, Faguet et al. and Kruger et al. noted that penicillin increased the activity of phages against *Staphylococci* when both antibacterial agents were administered simultaneously [66,67]. They also found that phages were less effective when given after penicillin, a phenomenon that could be explained by the fact that penicillin inhibits bacterial growth and therefore limits the activity of phages. In 2016, Torres-Barceló et al. showed that administering antibiotics after phages reduces bacterial density as well as bacterial resistance to either antibiotics or phages [68]. In 2018, Kumaran et al. confirmed that the effectiveness of phage–antibiotic combinations depends on the sequence of phage and antibiotic administration [69].

Akturk et al. evaluated the simultaneous or sequential combination of phages with seven antibiotics for the eradication of *S. aureus* and *P. aeruginosa* biofilms. They concluded that both the concentration of antibiotics and the sequence of phage and antibiotic administration are essential factors for effective phage–antibiotic therapy [52].

## 7. Mechanisms of Phage–Antibiotic Synergy

Several mechanisms of PAS have been described in the literature. The present review suggests that none of these mechanisms can alone explain the antibacterial effectiveness of phage–antibiotic combinations.

### 7.1. Recovery of Antibiotic Activity against Resistant Bacteria

Several studies have shown that phages can restore the antibacterial activity of antibiotics against resistant bacteria. In 1946, MacNeal et al. suggested that strains of *Staphylococci*, *Streptococci* and colon bacilli with moderate resistance to both penicillin and phages could be more effectively inhibited in their growth by combination of these antibacterial agents [17]. In 1962, Zilisteanu et al. found that the use of phages increased the in vitro sensitivity of two strains of *E. coli* and one strain of *Klebsiella* spp. to both chloromycetin and streptomycin [70]. They observed the same phenomenon in 1965 and 1969 with other bacterial strains [71,72]. In 2018, Chan et al. noted that the binding of phage OMKO1 to efflux pump proteins in a multi-resistant *P. aeruginosa* strain resulted in an evolutionary trade-off. As the killing of wildtype bacteria by phages increased bacterial resistance to phages, the bacterial sensitivity to ceftazidime or ciprofloxacin also increased [73]. A similar phenomenon was observed by Ho et al., who found that phage NPV1 caused strains of *E. faecalis* to develop NPV1-resistant mutations, which in turn made them more susceptible to daptomycin [74]. Lastly, Liu et al. suggested that phages could lower the minimal inhibition concentrations of resistant strains under certain conditions [75].

### 7.2. Action against Different Bacterial Sites

Manohar et al. and Torres-Barceló et al. showed that phage–antibiotic combinations can control bacterial proliferation and antibiotic resistance by targeting different bacterial receptors [68,76]. They explained this by the fact that it is more difficult for bacteria to develop resistance when they are invaded through different routes. This mechanism is similar to that observed with phage cocktails, which reduce the likelihood of emergence of multi-drug resistance by simultaneously attacking different bacterial receptors [77,78].

### 7.3. Changes in Bacterial Cell Morphology

In 2007, Comeau et al. noted that some antibiotics block bacterial cell division, causing bacterial cells to lengthen [32]. They argued that virulent phages take advantage of this bacterial elongation to increase their own replication, which increases their lytic activity against the infected bacteria and reinforces PAS [32]. In their 2013 study, Knezevic et al. noted that ceftriaxone caused the elongation of *P. aeruginosa* cells; however, they concluded that this change in cell morphology was a necessary but not sufficient reason for the observed PAS [40].

### 7.4. Antibiotic-Induced Phage Production

Antibiotics have been shown to induce phage production, which in turn reinforces the effect of PAS. Fothergill et al. examined the impact of six antibiotics on phage production in four isolates of a cystic fibrosis epidemic strain of *P. aeruginosa*. They found that the level of phage production varied according to both the antibiotic tested and the strain of *P. aeruginosa* [79]. Goerke et al. noted that the increase in phage production induced by the addition of ciprofloxacin resulted in enhanced recA transcription, which reflected the involvement of the SOS response of the bacteria [80]. Kim et al. suggested that increased phage production is the direct result of a lysis delay, itself caused by changes in bacterial cell morphology after the administration of sublethal concentrations of antibiotics [8].

### 7.5. Phage-Induced Penetration of Antibiotics into Biofilm

In the 2022 study by Necel et al., a phage cocktail combined with ciprofloxacin and rifampicin was able to successfully eradicate *E. coli* biofilm [62]. In 2020, Łusiak-Szelachowska et al. explained the efficacy of combined phage–antibiotic therapy against biofilm by the fact that PAS accelerates the degradation of the biofilm matrix by phage enzymes, which facilitates the penetration of antibiotics [81].

## 8. Future Perspectives

In 2021, Stachurska et al. proposed an in vitro method for detecting the PAS effect of different phage–antibiotic combinations that rests on the use of a unified double-layer agar [61].

Other perspectives for combined phage–antibiotic therapy involve augmenting the efficacy of phages. The interest in genome modification in this regard has been demonstrated by Lu et al., who found that modifying the genome of phages restored their antibacterial activity against *E. coli* [82]. Another interesting perspective is the use of endolysins—namely, phage enzymes that are produced late in the lytic cycle and that degrade the bacterial cell wall from the inside. In the future, endolysins could be synthesised or extracted from phages to be used directly as antibacterial agents [83]. Lastly, microencapsulation systems could be developed to help phages reach infected sites by slowing their degradation [84].

Future studies should explore the interactions between phages and the immune system, as high phage concentrations have been shown to induce a strong immune response, which in turn can reduce phage activity against bacteria [85].

## 9. Conclusions

Since the advent of combined phage–antibiotic therapy, several studies have explored the in vitro effect of PAS, and many interesting clinical cases have been reported. Our literature review suggests that phages as supplements to antibiotics can increase and/or prolong the antibacterial activity of antibiotics in a context of increasing bacterial resistance. Some questions, however, have yet to be resolved. Specific phage–antibiotic combinations should be tested to determine whether they are synergistic or antagonistic. The sequence of phage and antibiotic administration will need to be optimised for phage–antibiotic combinations to become routine therapeutic options. Future perspectives for combined phage–antibiotic therapy include gene modification, the use of endolysins and the development of microencapsulation systems.

## Figures and Tables

**Table 1 antibiotics-11-00839-t001:** Descriptions of phage–antibiotic combinations in the literature.

Year	Microorganism	Antibiotic	In Vivo	Human Infection	Observation	Reference
1941	*Staphyloccoci*	Sulphanilamide Sulphapyridine Sulphathiazole	No	No	Synergistic	[10]
1941	*S. aureus* *E. coli*	Sulphanilamide Sulphapyridine	No	No	Synergistic	[9]
1942	*Staphylococci*	Sulphathiazole	Rabbit	No	Synergistic	[11]
1943	*Staphylococci*	Neo-arsphenamine Sulphanilamide	/	Various types of infection	Synergistic	[12]
1943	*Staphylococci*	Neo-arsphenamine Sulphanilamide	/	Neurological infections	Synergistic	[13]
1945	*Staphylococci*	Penicillin	No	No	Synergistic	[14]
1945	*E. coli*	Streptothricin Streptomycin Clavacin	No	No	Antagonistic	[18]
1945	*Staphyloccoci*	Penicillin	/	Cardiac infections	Synergistic	[15]
1945	*Staphyloccoci* *Streptococci*	Penicillin	/	Cardiac infections	Synergistic	[16]
1946	*Staphyloccoci* *Streptococci* *E. coli*	Penicillin	/	Various types of infection	Synergistic	[17]
1949 1950	*S. aureus* *E. coli*	Streptomycin Chloromycetin Aureomycitin Terramycin	No	No	Antagonistic	[19,20,21,22,23,24,25]
1951	Several bacteria	Quinones Clavacin Subtilin Penicillic acid Ploymixin B Streptomycin	No	No	Antagonistic	[26]
1954	*S. aureus*	Aureomycin Oxytetracyclin Chloramphenicol	No	No	Antagonistic	[27]
1965	*Dysenteric bacilli*	Chloramphenicol Tetracycline	/	Digestive colonisation	Synergistic	[28]
1971	Enterobacteriae *Staphylococci*	Not specified	No	No	Synergistic	[29]
1977	*E. coli*	Rifampicin	No	No	Antagonistic	[30]
2006	*P. aeruginosa*	β-Lactam Chloramphenicol Gentamycin	Mice	No	Synergistic	[31]
2007	*E. coli*	Β-Lactam Quinolone	No	No	Synergistic	[32]
2008	*P. aeruginosa*	Β-Lactam Quinolone Aminoglycosids	No	No	Synergistic	[33]
2008	*Salmonella enterica*	Gentamicin Ciprofloxacin	No	No	Synergistic	[34]
2009	*K.* *pneumoniae*	Amoxicillin	No	No	Synergistic	[35]
2011	*S. aureus*	Rifampicin	No	No	Synergistic	[36]
2012	*S. aureus*	Gentamycin	No	No	Synergistic	[37]
2012	*E. coli*	Cefotaxime	No	No	Synergistic	[38]
2013	*S. aureus*	Linezolid	No	No	Synergistic	[39]
2013	*P. aeruginosa*	Ceftriaxone	No	No	Synergistic	[40]
2013	*S. aureus* *P. aeruginosa*	Teicoplanin Imipenem Amikacin	Rat	No	Synergistic	[41]
2014	*Mycobacterium smegmatis*	Ethambutol	No	No	Synergistic	[42]
2014	*E. coli* *P. aeruginosa*	Tobramycin	No	No	Synergistic	[43]
2014	*S. aureus*	Linezolid	No	No	Synergistic	[44]
2014	*P. aeruginosa*	Streptomycin	No	No	Synergistic	[45]
2015	*S. aureus*	Gentamycin Vancomycin Tetracyclin	No	No	Synergistic	[46]
2015	*Burkholderia cepacia*	Ciprofloxacin Meropenem Tetracycline Minocycline Levofloxacin Ceftazidime	No	No	Synergistic	[47]
2016	*S. aureus*	Linezolid	Mice	No	Synergistic	[48]
2017	*P. aeruginosa*	Ceftazidime Ciprofloxacin Tobramycin	No	No	Synergistic	[49]
2018	*P. aeruginosa*	Ciprofloxacin Tobramycin Colistin Aztreonam Amikacin	No	No	Synergistic	[50]
2018	*P. aeruginosa*	Piperacillin Ceftazidime	No	No	Synergistic	[51]
2019	*P. aeruginosa* *S. aureus*	Aminoglycosid Tetracycline Chloramphenicol Meropenem Erythromycin Ciprofloxacin	No	No	Synergistic	[52]
2019	*Enterococci*	Vancomycin	No	No	Synergistic	[53]
2020	*S. aureus*	Dalbavancin	/	Neurological infection	Synergistic	[54]
2020	*S. aureus*	Daptomycin	/	Osteoarticular infection	Clinical failure	[55]
2020	*Achromobacter* spp.	Cefiderocol Meropenem-varobactam	/	Pulmonary infection	Synergistic	[56]
2020	*Acinetobacter baumanii*	Ciprofloxacin Meropenem	No	No	Synergistic	[57]
2020	*Enterococcus faecium*	Daptomycin	No	No	Synergistic	[58]
2020	*Serratia marcescens*	Ampicillin- Sulbactam	No	No	Synergistic	[59]
2021	*E. coli* *Bacillus cereus*	Aminoglycosids	No	No	Antagonistic	[60]
2021	*E. coli*	β-lactams Fluoroquinolones Colistin sulphate Trimethoprim-sulphamethoxazole Tetracyclines Amino-glycosides	No	No	Synergistic Possibly antagonistic	[61]
2022	*E. coli*	Ciprofloxacin Rifampicin	No	No	Synergistic	[62]

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
