# Peer review of "Benefits of Combined Phage–Antibiotic Therapy for the Control of Antibiotic-Resistant Bacteria: A Literature Review"

_antibiotics, 2022, doi:10.3390/antibiotics11070839_

Round 1
Reviewer 1 Report
Dear Authors
I have read your review article "Benefits of Combined Phage-Antibiotic Therapy for Controlling Antibiotic-Resistant Bacteria" and have the following comments on it:
- The review article does not provide a brief description of PAS, as I can see in the literature there are so many reports on PAS which are not included here. I suggest authors to add a table of PAS with their outcomes.
- Section "Examples of associations described" is very superficial and must be written in detail with respect to different potent pathogens.
- Authors may create a figure for mechanisms of PAS action, for better visualization.
- Sections 5.6, 5.7 and 6 contains not have much information and need to be rewritten in detail.
- I suggest authors to add future directions such as the use of encapsulation systems, endolysins etc to increase the therapeutic efficacy of phages.
- Immunologic reaction or Immunological reactions? This section needs to be rewritten in detail.
- Section 8: Patents? I cannot understand this.
- In conclusion, this review in its present form cannot be considered for publication. I suggest authors to rewrite the suggested sections in detail with appropriate tables and figures.
Author Response
Reviewer 1
I have read your review article "Benefits of Combined Phage-Antibiotic Therapy for Controlling Antibiotic-Resistant Bacteria" and have the following comments on it:
The review article does not provide a brief description of PAS, as I can see in the literature there are so many reports on PAS which are not included here. I suggest authors to add a table of PAS with their outcomes.
We added the Table 1. Description of phage-antibiotic synergy in the literature.
Section "Examples of associations described" is very superficial and must be written in detail with respect to different potent pathogens.
We synthetised the associations described in Table 1. We have also restructured the paragraphs and provided additional information on the various studies.
Authors may create a figure for mechanisms of PAS action, for better visualization.
Since several mechanisms have been described and several are supposed, it is difficult to create a figure to synthesize all the proposed scenarios.
Sections 5.6, 5.7 and 6 contains not have much information and need to be rewritten in detail.
We have rewritten and restructured the paragraphs by providing details on the various studies.
I suggest authors to add future directions such as the use of encapsulation systems, endolysins etc to increase the therapeutic efficacy of phages.
We have changed the structure of paragraph 6 and added the following sentence : “Future perspectives for combined phage-antibiotic therapy include gene modification and the use of endolysins and encapsulation systems.”
Immunologic reaction or Immunological reactions? This section needs to be rewritten in detail.
We removed this paragraph, and we have modified this paragraph by providing additional information on the study in question: “Future studies should explore the interactions between phages and the immune system, as high phage concentrations have been shown to induce a strong immune response, which in turn can reduce phage activity against bacteria”.
Section 8: Patents? I cannot understand this.
This section has been removed.
In conclusion, this review in its present form cannot be considered for publication. I suggest authors to rewrite the suggested sections in detail with appropriate tables and figures.

Reviewer 2 Report
The review by Kevin Diallo and Alain Dublanchet discusses combination of phage therapy and antibiotics to treat bacterial infections.
- The review requires extensive English language editing. The authors must get the manuscript checked by professional language editors before submitting the manuscript.
- The review does is not structured very well and makes the reading difficult
- The review does not add anything significant to the currently available literature.
- The authors only enlist different studies but do not synthesize from them.
- Overall the review is not well written and cant be accepted for publication in its current form
Author Response
Reviewer 2
The review by Kevin Diallo and Alain Dublanchet discusses combination of phage therapy and antibiotics to treat bacterial infections.
The review requires extensive English language editing. The authors must get the manuscript checked by professional language editors before submitting the manuscript.
The manuscript was proofread by an English speaker for the new submission.
The review does is not structured very well and makes the reading difficult
We have rewritten and restructured the paragraphs by providing details on the various studies.
The review does not add anything significant to the currently available literature.
We have tried to synthesize all the clinical uses of phage therapy associated with antibiotic therapy, in order to demonstrate its effectiveness. We have added Table 1 with the outcome of these uses.
The authors only enlist different studies but do not synthesize from them.
Table 1 considers the outcome of these studies.
Overall the review is not well written and cant be accepted for publication in its current form

Reviewer 3 Report
The manuscript : “Benefits of Combined Phage-Antibiotic Therapy for Controlling Antibiotic Resistant Bacteria” by Diallo and Dublanchet is relative clear and informative. However, it still contains several points that need to be improved.
Major comment: Please check grammar mistake and improve academic English in whole manuscript.
My minor comments as stated below:
Line 26: Some (several): should be several
The introduction needs to add a sentence that mentions your review's aims/ what you want to give to the reader. The current form, the introduction could not indicate what you want to say.
Line 46-47: please revise your English. Repeated phrases/words occur many times in this sentence: “with, and”
Line 58-59: “Complete lysis occurred in 6 hours and 3 minutes for penicillin alone (a), in 1 hour and 55 minutes for phage alone (b), and in 1 hour and 25 minutes for both (c)” This sentence you need to describe more detail, otherwise reader could not understand what are you want to say, for example why or how in different conditions for 6 hours/3 minutes for penicillin alone?
Line 71: repeated 3 “and” in a sentence, please use comma instead of “and”
Line 76: In vitro should be italic
Line 88: “Chhibber 87 et al. continued in 2013 his research on….” What is his previous research? Please rewrite this sentence.
Section 3.2: the writing is not good enough. Please create the graph with the topic sentence and connection between sentences and check the writing. I think it should be a table instead of the list of each author with a result and separate each sentence in its current form.
Line 119: [35] should be moved to the end of the sentence
Line 146-148: so much “to” in this sentence. Writing is not good enough.
Several Latin names of bacteria should be italic, such as “P. aeruginosa” line 236
Author Response
Reviewer 3
The manuscript : “Benefits of Combined Phage-Antibiotic Therapy for Controlling Antibiotic Resistant Bacteria” by Diallo and Dublanchet is relative clear and informative. However, it still contains several points that need to be improved.
Major comment: Please check grammar mistake and improve academic English in whole manuscript.
The manuscript was proofread by an English speaker for the new submission.
My minor comments as stated below:
Line 26: Some (several): should be several
We rewrote the introduction to be more understandable.
The introduction needs to add a sentence that mentions your review's aims/ what you want to give to the reader. The current form, the introduction could not indicate what you want to say.
We added the following sentence : “The aim of this literature review was to provide an overview of phage-antibiotic combinations and to describe the possible mechanisms of PAS”.
Line 46-47: please revise your English. Repeated phrases/words occur many times in this sentence: “with, and”
The manuscript was proofread by an English speaker for the new submission.
Line 58-59: “Complete lysis occurred in 6 hours and 3 minutes for penicillin alone (a), in 1 hour and 55 minutes for phage alone (b), and in 1 hour and 25 minutes for both (c)” This sentence you need to describe more detail, otherwise reader could not understand what are you want to say, for example why or how in different conditions for 6 hours/3 minutes for penicillin alone?
We rewrote the sentence to be more understandable: “In 1945, Himmelweit et al. broth cultured 130 million bacteria/mL of Staphylococci, to which they added (a) penicillin, (b) phages and (c) a combination of both. They found that complete lysis occurred in 6 hours and 3 minutes for penicillin alone (a), in 1 hour and 55 minutes for phages alone (b) and in 1 hour and 25 minutes for the phage-penicillin com-bination (c).”
Line 71: repeated 3 “and” in a sentence, please use comma instead of “and”
The manuscript was proofread by an English speaker for the new submission.
Line 76: In vitro should be italic
We have italicized the words in vivo and in vitro in the manuscript.
Line 88: “Chhibber 87 et al. continued in 2013 his research on….” What is his previous research? Please rewrite this sentence.
Lines 79 and 80: We rewrote the sentence to be more understandable: “In 2013, Chhibber et al. highlighted the synergistic effect of combining phages with linezolid against methicillin-resistant strains of S. aureus”
Section 3.2: the writing is not good enough. Please create the graph with the topic sentence and connection between sentences and check the writing. I think it should be a table instead of the list of each author with a result and separate each sentence in its current form.
The manuscript was proofread by an English speaker for the new submission, and we have added table 1 which summarizes the different uses.
Line 119: [35] should be moved to the end of the sentence
We moved the reference as suggested.
Line 146-148: so much “to” in this sentence. Writing is not good enough.
The manuscript was proofread by an English speaker for the new submission.
Several Latin names of bacteria should be italic, such as “P. aeruginosa” line 236
We checked the document to put the name of all microorganisms in italics as suggested.

Round 2
Reviewer 1 Report
Authors have extensively worked on the improvement of the manuscript but authors must understand that this paper is being considered for full review, not a opinion or mini-review. Despite many changes I still have the following issues:
1. Table 1: author's name should not be there, only the year of publication is required. The in-vivo model system and infection model should be added for clarity. The outcome is also not defined in a scientific manner. Table legends are also inconsistent. Some of the organisms name are full where as some are abbreviated, it should be full whereever appears first.
2. Mechanism of phage-antibiotic synergy section is not well written. It should be written in a more detailed manner. For example section 7.5.
3. Future perspectives are very superficial and seem to be written only to respond to reviewers. This should be written with consideration to lysins, delivery systems, their combinations.
4. Introduction is still very brief, the authors need to discuss it in detail to make a better platform for readers.
5. Section 3: Organisms names are not depicted scientifically correct.
Author Response
Authors have extensively worked on the improvement of the manuscript but authors must understand that this paper is being considered for full review, not a opinion or mini-review. Despite many changes I still have the following issues:
- Table 1: author's name should not be there, only the year of publication is required. The in-vivo model system and infection model should be added for clarity. The outcome is also not defined in a scientific manner. Table legends are also inconsistent. Some of the organisms name are full where as some are abbreviated, it should be full whereever appears first.
We have removed the column naming the authors. We specified the in vitro and clinical models used. We changed "outcome" to "observation" in the table, and defined by introducing the table at line 49: “Observed results were synergistic if the combination of phage and antibiotic was more effective than phages alone or antibiotics alone ; results were antagonistic if the combination of phage and antibiotics worked worse than phage alone or phage alone. In the case of an in vivo study, the number of clinical successes was reported.” We have changed the names of organisms so that they appear complete wherever they appear first.
- Mechanism of phage-antibiotic synergy section is not well written. It should be written in a more detailed manner. For example section 7.5.
7.2. We added the sentences: “Some phages are specific for one or a few strains of a bacterial species. It is often necessary to use a mixture of phages or a cocktail of phages to target genetically different bacteria on the same infected site. Phage cocktails prevent the emergence of bacterial resistance to phages, because it is difficult for a bacterium to develop resistance to several phages.”
7.3. We added the sentences: “The virulent phages can take advantage of this damage to the bacteria to increase their own production and the destruction of the infected bacteria also allows the phages to spread more quickly. Moreover, the bacterium infected by the phage has a metabolism modified and conditioned by the genome of the phage.”
7.5. We added the following sentence: “Indeed, phages can be used with other antimicrobials to improve the efficiency of biofilm removal, allow synergy through mechanical debridement and using enzymes such as depolymerases which are enzymes produced by the phage to destroy the biofilm matrix.”
- Future perspectives are very superficial and seem to be written only to respond to reviewers. This should be written with consideration to lysins, delivery systems, their combinations.
We have specified the different perspectives to come by defining the tools that could be used:
“Endolysins are peptidoglycan-degrading enzymes synthesized late in the lytic cycle, to de-grade the bacterial cell wall from the inside. Several studies have been conducted to use these substances with antibacterial activity. Microencapsulation is defined by the packaging of materials in capsules with an encapsulation material chosen so that the release of the encapsulated element can be achieved under certain conditions. The goal would be to be able to bring the phages without degrading them to the infected site.”
- Introduction is still very brief, the authors need to discuss it in detail to make a better platform for readers.
We provided more information about phage therapy: “Phages (or bacteriophages) are bacteria-specific viruses that use the bacteria's metabolism to replicate. Lytic phages allow the death of the bacteria they infect. Phages are the most abundant microorganisms in the biosphere and are present in all areas where bacteria grow and play an important role in ecosystems. It is accepted that independently discovered by Frederick Twort, as ultra-microscopic viruses in 1915 and named by Félix d'Hérelle in 1917. Phage therapy has several advantages: the phages self-replicate; a greater specificity than antibiotics; a low rate of side effects; they are an option for allergic patients; they have low production costs; different routes of administration are possible..”
- Section 3: Organisms names are not depicted scientifically correct.
We have made the following clarifications or changes to the names:
Lines 100 and 101: “with the filamentous P. aeruginosa phage Pf3 and P. aeruginosa phage Pf1”
Line 115: “the treatment of infections due to E. coli;”
Line 119: “Acinetobacter baumanii,”
Line 123: “Mycobacteriophage D29”

Reviewer 2 Report
Authors have incorporated all the suggestions from the previous draft
Author Response
Authors have incorporated all the suggestions from the previous draft.
We thank reviewer 2 for his comments which helped to improve the manuscript.

Round 3
Reviewer 1 Report
The authors have made some changes according to my suggestions but I still have some questions/suggestions:
1. The introduction is still weak, short and lacking references. I can see only two references for the whole introduction.
2. Are there only two reports in France for the compassionate use of phage therapy? Authors must give details of those reports with outcomes and look for some other unpublished reports in the same context.
3. "The sequence of phage and antibiotic administration" title is misleading and must be replaced.
4. Section 7.2: Do you mean Action against different bacterial strains? Please check the 7.2 heading.
5. Section 7.5 is still very concise and the authors must discuss it in more detail with possible mechanisms of phage penetration along with references.
6. The added data in future perspectives is not clear. Authors have defined a major antibacterial agent endolysins in 1-2 lines without any reference, which is required to be discussed in detail with references. While talking about encapsulation authors must write on the encapsulation of phages and endolysins rather than the general encapsulation.
In my point of view, authors must give substantial efforts to improving the quality of the present manuscript.
Author Response
Editor-in-Chief
Antibiotics
17th June 2022
Dear Editor,
We thank you for giving us the opportunity to revise our manuscript entitled “Benefits of combined phage-antibiotic therapy for the control of antibiotic-resistant bacteria: A literature review”. Attached please find the revised version of this manuscript.
We addressed all the Editor’s comments.
The authors have introduced many points suggested in previous reports of reviewers. However, this manuscript still suffers from some weaknesses. Following point should be addressed:
- English usage is still not acceptable, especially in the text marked in yellow background.
The manuscript was revised by a professional medical translator.
Please note that we were not sure as to why some terms in Table 1 were marked in yellow. We interpreted this as meaning that the use of terms had to be standardised. As we proceeded with this standardisation, we decided to explain in sections 3 and 4 why combinations were classified as synergistic or antagonistic.
- Some statements are simply not true. For example «Lytic phages allow the death of the bacteria they infect" (lines 31-32) does not make sense. This sentence should be replaced by «Lytic phages cause the death of bacteria they infect". There is a fundamental difference between allowing for something and causing something.
We have rewritten the sentence at the request of the editor.
- Introduction is extremely poorly referenced. Just two papers cited in the introductory chapter is definitely not appropriate in a review article. Please, cite previously published works when presenting previous discoveries and current facts.
We have added several references to our introduction as suggested.
- Developmental styles of bacteriophages should be presented briefly, and parasite-host relationship between phages and bacteria should be described shortly, in order to allow a general reader to understand basic features of phage biology.
We have described the developmental styles of phages and detailed the mechanisms of infection of bacteria by phages in the Introduction.
- Still, recent literature on combined phage-antibiotic therapy is presented very selectively. Even a special issue of this journal, entitled ""Antibiotics vs. Phage Therapy" (https://www.mdpi.com/journal/antibiotics/special_issues/anti_phage) has been ignored, and articles published in this issue were not even mentioned. At least, those related to combination of these two types of antibacterial action should be discussed briefly and included in Table 1.
As suggested by the editor, we have discussed briefly and added more recent references to Table 1.
All authors have seen and approved this updated version of the manuscript. They have all contributed significantly to the work. None of the authors have any conflicts of interest to disclose.
Thank you for evaluating our revised manuscript.
Yours sincerely,
This manuscript is a resubmission of an earlier submission. The following is a list of the peer review reports and author responses from that submission.
Round 1
Reviewer 1 Report
The manuscript “Benefits of Combined Phage-Antibiotic Therapy for Controlling Antibiotic Resistant Bacteria” provides a review citing various phage-antibiotic studies in the literature. The manuscript is well written and organized.
Major comments:
All topics and sub-topics are limited to listing the studies done, and the discussion (if present) remains superficial. A more overarching view in 6. Perspective section on the future prospects of the combined use of phages and antibiotics in therapy is welcome. I would like to read more about the author’s vision in all topics addressed in this review. More information/authors opinion whether the antibiotic and phage synergies and antagonisms are directly associated with the type of phage family/genus, the bacterial host, and mechanism of action of antibiotics would be welcome. What can be done to minimize antagonism? Sometimes an additive effect can be reached, which is better than using phages and antibiotics alone, so has this been observed for P. aeruginosa or S. aureus with some phage-antibiotic combinations?
I understand that many of the works only report synergism or antagonism, but recent works possibly provide an answer or at least a hypothesis (e.g., lines 79-80 provide the reason behind the antagonism observed; throughout 3. Examples of associations described) that can complement the limited information provided in 5.1. and 5.3.
Provide further information on 5.7. on how do phages degrade the matrix. The sentence included only reports to the action of phages, but what about the combined antibiotic-phage interaction with biofilm bacteria and matrix?
Minor comments:
The title "4. Pathologies" does not fit well to the description of the sub-topics provided. Please provide a better title for the section.
Reviewer 2 Report
This paper is a literature review of the interaction between phages and antibiotics in the optic of human treatment against bacterial infection. The review is consistent overall, but I have some comments:
- Words in Latin such as in vivo, in vitro, et al., need to be italicized.
- Sections 3 through the end are well written, but the English need to be heavily revised in sections 1 and 2. (example: use of present tense instead of past, was used instead of were, wordy sentences…)
- The words in between parenthesis in the introductions are strange and should be removed.
- At the end of section 2 (lines 79-80), the last statement lack clarity and should be further explained.
- In sections 3 and 4, the authors state these parts have been reviewed by others a few years before. The authors should also point at what is similar and different in their review.
- Section 8 is empty and should be removed or filled.